# DISAGREEMENT-REGULARIZED IMITATION LEARNING

**Kianté Brantley** *
University of Maryland
kdbrant@cs.umd.edu

**Wen Sun**
Microsoft Research
sun.wen@microsoft.com

**Mikael Henaff**
Microsoft Research
mihenaff@microsoft.com

## ABSTRACT

We present a simple and effective algorithm designed to address the covariate shift problem in imitation learning. It operates by training an ensemble of policies on the expert demonstration data, and using the variance of their predictions as a cost which is minimized with RL together with a supervised behavioral cloning cost. Unlike adversarial imitation methods, it uses a fixed reward function which is easy to optimize. We prove a regret bound for the algorithm which is linear in the time horizon multiplied by a coefficient which we show to be low for certain problems on which behavioral cloning fails. We evaluate our algorithm empirically across multiple pixel-based Atari environments and continuous control tasks, and show that it matches or significantly outperforms behavioral cloning and generative adversarial imitation learning.

## 1 INTRODUCTION

Training artificial agents to perform complex tasks is essential for many applications in robotics, video games and dialogue. If success on the task can be accurately described using a reward or cost function, reinforcement learning (RL) methods offer an approach to learning policies which has proven to be successful in a wide variety of applications (Mnih et al., 2015; 2016; Lillicrap et al., 2016; Hessel et al., 2018). However, in other cases the desired behavior may only be roughly specified and it is unclear how to design a reward function to characterize it. For example, training a video game agent to adopt more human-like behavior using RL would require designing a reward function which characterizes behaviors as more or less human-like, which is difficult.

Imitation learning (IL) offers an elegant approach whereby agents are trained to mimic the demonstrations of an expert rather than optimizing a reward function. Its simplest form consists of training a policy to predict the expert's actions from states in the demonstration data using supervised learning. While appealingly simple, this approach suffers from the fact that the distribution over states observed at execution time can differ from the distribution observed during training. Minor errors which initially produce small deviations become magnified as the policy encounters states further and further from its training distribution. This phenomenon, initially noted in the early work of (Pomerleau, 1989), was formalized in the work of (Ross & Bagnell, 2010) who proved a quadratic $\mathcal{O}(\epsilon T^2)$ bound on the regret and showed that this bound is tight. The subsequent work of (Ross et al., 2011) showed that if the policy is allowed to further interact with the environment and make queries to the expert policy, it is possible to obtain a *linear* bound on the regret. However, the ability to query an expert can often be a strong assumption.

In this work, we propose a new and simple algorithm called DRIL (Disagreement-Regularized Imitation Learning) to address the covariate shift problem in imitation learning, in the setting where the agent is allowed to interact with its environment. Importantly, the algorithm does not require any additional interaction with the expert. It operates by training an ensemble of policies on the demonstration data, and using the disagreement in their predictions as a cost which is optimized through RL together with a supervised behavioral cloning cost. The motivation is that the policies in the ensemble will tend to agree on the set of states covered by the expert, leading to low cost, but are more likely to disagree on states not covered by the expert, leading to high cost. The RL cost

---

*Work done while at Microsoft Research.

thus guides the agent back towards the distribution of the expert, while the supervised cost ensures that it mimics the expert within the expert's distribution.

Our theoretical results show that, subject to realizability and optimization oracle assumptions[1], our algorithm obtains a $\mathcal{O}(\epsilon \kappa T)$ regret bound, where $\kappa$ is a measure which quantifies a tradeoff between the concentration of the demonstration data and the diversity of the ensemble outside the demonstration data. We evaluate DRIL empirically across multiple pixel-based Atari environments and continuous control tasks, and show that it matches or significantly outperforms behavioral cloning and generative adversarial imitation learning, often recovering expert performance with only a few trajectories.

## 2 PRELIMINARIES

We consider episodic finite horizon MDP in this work. Denote by $\mathcal{S}$ the state space, $\mathcal{A}$ the action space, and $\Pi$ the class of policies the learner is considering. Let $T$ denote the task horizon and $\pi^\star$ the expert policy whose behavior the learner is trying to mimic. For any policy $\pi$, let $d_\pi$ denote the distribution over states induced by following $\pi$. Denote $C(s, a)$ the expected immediate cost of performing action $a$ in state $s$, which we assume is bounded in $[0, 1]$. In the imitation learning setting, we do not necessarily know the true costs $C(s, a)$, and instead we observe expert demonstrations. Our goal is to find a policy $\pi$ which minimizes an observed surrogate loss $\ell$ between its actions and the actions of the expert under its induced distribution of states, i.e.

$$\hat{\pi} = \arg\min \mathbb{E}_{s \sim d_\pi}[\ell(\pi(s), \pi^\star(s))] \tag{1}$$

For the following, we will assume $\ell$ is the total variation distance (denoted by $\|\cdot\|$), which is an upper bound on the $0 - 1$ loss. Our goal is thus to minimize the following quantity, which represents the distance between the actions taken by our policy $\pi$ and the expert policy $\pi^\star$:

$$J_{\exp}(\pi) = \mathbb{E}_{s \sim d_\pi}\left[\|\pi(\cdot|s) - \pi^\star(\cdot|s)\|\right] \tag{2}$$

Denote $Q_t^\pi(s, a)$ as the standard Q-function of the policy $\pi$, which is defined as $Q_t^\pi(s, a) = \mathbb{E}\left[\sum_{\tau=t}^{T} C(s_\tau, a_\tau)|(s_t, a_t) = (s, a), a_\tau \sim \pi\right]$. The following result shows that if $\ell$ is an upper bound on the $0 - 1$ loss and $C$ satisfies certain smoothness conditions, then minimizing this loss within $\epsilon$ translates into an $\mathcal{O}(\epsilon T)$ regret bound on the true task cost $J_C(\pi) = \mathbb{E}_{s, a \sim d_\pi}[C(s, a)]$:

**Theorem 1.** *(Ross et al., 2011) If $\pi$ satisfies $J_{\exp}(\pi) = \epsilon$, and $Q_{T-t+1}^{\pi^\star}(s, a) - Q_{T-t+1}^{\pi^\star}(s, \pi^\star) \leq u$ for all time steps $t$, actions $a$ and states $s$ reachable by $\pi$, then $J_C(\pi) \leq J_C(\pi^\star) + uT\epsilon$.*

Unfortunately, it is often not possible to optimize $J_{\exp}$ directly, since it requires evaluating the expert policy on the states induced by following the *current* policy. The supervised behavioral cloning cost $J_{BC}$, which is computed on states induced by the expert, is often used instead:

$$J_{BC}(\pi) = \mathbb{E}_{s \sim d_{\pi^\star}}[\|\pi^\star(\cdot|s) - \pi(\cdot|s)\|] \tag{3}$$

Minimizing this loss within $\epsilon$ yields a quadratic regret bound on regret:

**Theorem 2.** *(Ross & Bagnell, 2010) Let $J_{BC}(\pi) = \epsilon$, then $J_C(\pi) \leq J_C(\pi^\star) + T^2\epsilon$.*

Furthermore, this bound is tight: as we will discuss later, there exist simple problems which match the worst-case lower bound.

## 3 ALGORITHM

Our algorithm is motivated by two criteria: i) the policy should act similarly to the expert within the expert's data distribution, and ii) the policy should move towards the expert's data distribution

---

[1]We assume for the analysis the action space is discrete, but the state space can be large or infinite.

---

**Algorithm 1** Disagreement-Regularized Imitation Learning (DRIL)

---
1: **Input:** Expert demonstration data $\mathcal{D} = \{(s_i, a_i)\}_{i=1}^N$
2: Initialize policy $\pi$ and policy ensemble $\Pi_E = \{\pi_1, ..., \pi_E\}$
3: **for** $e = 1, E$ **do**
4:     Sample $\mathcal{D}_e \sim \mathcal{D}$ with replacement, with $|\mathcal{D}_e| = |\mathcal{D}|$.
5:     Train $\pi_e$ to minimize $J_{\mathrm{BC}}(\pi_e)$ on $\mathcal{D}_e$ to convergence.
6: **end for**
7: **for** $i = 1, ...$ **do**
8:     Perform one gradient update to minimize $J_{\mathrm{BC}}(\pi)$ using a minibatch from $\mathcal{D}$.
9:     Perform one step of policy gradient to minimize $\mathbb{E}_{s \sim d_\pi, a \sim \pi(\cdot|s)}[C_{\mathrm{U}}^{\mathrm{clip}}(s, a)]$.
10: **end for**

---

if it is outside of it. These two criteria are addressed by combining two losses: a standard behavior cloning loss, and an additional loss which represents the variance over the outputs of an ensemble $\Pi_E = \{\pi_1, ..., \pi_E\}$ of policies trained on the demonstration data $\mathcal{D}$. We call this the uncertainty cost, which is defined as:

$$C_{\mathrm{U}}(s, a) = \mathrm{Var}_{\pi \sim \Pi_E}(\pi(a|s)) = \frac{1}{E} \sum_{i=1}^E \left( \pi_i(a|s) - \frac{1}{E} \sum_{i=1}^E \pi_i(a|s) \right)^2$$

The motivation is that the variance over plausible policies is high outside the expert's distribution, since the data is sparse, but it is low inside the expert's distribution, since the data there is dense. Minimizing this cost encourages the policy to return to regions of dense coverage by the expert. Intuitively, this is what we would expect the expert policy $\pi^\star$ to do as well. The total cost which the algorithm optimizes is given by:

$$J_{\mathrm{alg}}(\pi) = \underbrace{\mathbb{E}_{s \sim d_{\pi^\star}}[\|\pi^\star(\cdot|s) - \pi(\cdot|s)\|]}_{J_{\mathrm{BC}}(\pi)} + \underbrace{\mathbb{E}_{s \sim d_\pi, a \sim \pi(\cdot|s)}[C_{\mathrm{U}}(s, a)]}_{J_{\mathrm{U}}(\pi)}$$

The first term is a behavior cloning loss and is computed over states generated by the expert policy, of which the demonstration data $\mathcal{D}$ is a representative sample. The second term is computed over the distribution of states generated by the current policy and can be optimized using policy gradient.

Note that the demonstration data is fixed, and this ensemble can be trained once offline. We then interleave the supervised behavioral cloning updates and the policy gradient updates which minimize the variance of the ensemble. The full algorithm is shown in Algorithm 1. We also found that dropout (Srivastava et al., 2014), which has been proposed as an approximate form of ensembling, worked well (see Appendix D).

In practice, for the supervised loss we optimize the KL divergence between the actions predicted by the policy and the expert actions, which is an upper bound on the total variation distance due to Pinsker's inequality. We also found it helpful to use a clipped uncertainty cost:

$$C_{\mathrm{U}}^{\mathrm{clip}}(s, a) = \begin{cases} -1 & \text{if } C_{\mathrm{U}}(s, a) \leq q \\ +1 & \text{else} \end{cases}$$

where the threshold $q$ is a top quantile of the raw uncertainty costs computed over the demonstration data. The threshold $q$ defines a normal range of uncertainty based on the demonstration data, and values above this range incur a positive cost (or negative reward).

The RL cost can be optimized using any policy gradient method. In our experiments we used advantage actor-critic (A2C) (Mnih et al., 2016) or PPO (Schulman et al., 2017), which estimate the expected cost using rollouts from multiple parallel actors all sharing the same policy (see Appendix C for details). We note that model-based RL methods could in principle be used as well if sample efficiency is a constraint.

## 4 ANALYSIS

### 4.1 COVERAGE COEFFICIENT

We now analyze DRIL for MDPs with discrete action spaces and potentially large or infinite state spaces. We will show that, subject to assumptions that the policy class contains an optimal policy and that we are able to optimize costs within $\epsilon$ of their global minimum, our algorithm obtains a regret bound which is linear in $\kappa T$, where $\kappa$ is a quantity which depends on the environment dynamics, the expert distribution $d_\pi^\star$, and our learned ensemble. Intuitively, $\kappa$ represents a tradeoff between how concentrated the demonstration data is and how high the variance of the ensemble is outside the expert distribution.

**Assumption 1.** *(Realizability)* $\pi^\star \in \Pi$.

**Assumption 2.** *(Optimization Oracle) For any given cost function $J$, our minimization procedure returns a policy $\hat{\pi} \in \Pi$ such that $J(\hat{\pi}) \leq \arg\min_{\pi \in \Pi} J(\pi) + \epsilon$.*

The motivation behind our algorithm is that the policies in the ensemble agree inside the expert's distribution and disagree outside of it. This defines a reward function which pushes the learner back towards the expert's distribution if it strays away. However, what constitutes inside and outside the distribution, or sufficient agreement or disagreement, is ambiguous. Below we introduce quantities which makes these ideas precise.

**Definition 1.** *For any set $\mathcal{U} \subseteq \mathcal{S}$, define the concentrability inside of $\mathcal{U}$ as $\alpha(\mathcal{U}) = \max_{\pi \in \Pi} \sup_{s \in \mathcal{U}} \frac{d_\pi(s)}{d_{\pi^\star}(s)}$.*

The notion of concentrability has been previously used to give bounds on the performance of value iteration (Munos & Szepesvári, 2008). For a set $\mathcal{U}$, $\alpha(\mathcal{U})$ will be low if the expert distribution has high mass at the states in $\mathcal{U}$ that are reachable by policies in the policy class.

**Definition 2.** *Define the minimum variance of the ensemble outside of $\mathcal{U}$ as $\beta(\mathcal{U}) = \min_{s \notin \mathcal{U}, a \in \mathcal{A}} \mathrm{Var}_{\pi \sim \Pi_E}[\pi(a|s)]$.*

We now define the $\kappa$ coefficient as the minimum ratio of these two quantities over all possible subsets of $\mathcal{S}$.

**Definition 3.** *We define $\kappa = \min_{\mathcal{U} \subseteq \mathcal{S}} \frac{\alpha(\mathcal{U})}{\beta(\mathcal{U})}$.*

We can view $\kappa$ as the quantity which minimizes the tradeoff over different subsets $\mathcal{U}$ between coverage by the expert policy inside of $\mathcal{U}$, and variance of the ensemble outside of $\mathcal{U}$.

### 4.2 REGRET BOUND

We now establish a relationship between the $\kappa$ coefficient just defined, the cost our algorithm optimizes, and $J_{\mathrm{exp}}$ defined in Equation (2) which we would ideally like to minimize and which translates into a regret bound. All proofs can be found in Appendix A.

**Lemma 1.** *For any $\pi \in \Pi$, we have $J_{\mathrm{exp}}(\pi) \leq \kappa J_{\mathrm{alg}}(\pi)$.*

This result shows that if $\kappa$ is not too large, and we are able to make our cost function $J_{\mathrm{alg}}(\pi)$ small, then we can ensure $J_{\mathrm{exp}}(\pi)$ is also small. This result is only useful if our cost function can indeed achieve a small minimum. The next lemma shows that this is the case.

**Lemma 2.** $\min_{\pi \in \Pi} J_{\mathrm{alg}}(\pi) \leq 2\epsilon$.

Here $\epsilon$ is the threshold specified in Assumption 2. Combining these two lemmas with the previous result of Ross et al. (2011), we get a regret bound which is linear in $\kappa T$.

**Theorem 3.** *Let $\hat{\pi}$ be the result of minimizing $J_{\mathrm{alg}}$ using our optimization oracle, and assume that $Q_{T-t+1}^{\pi^\star}(s, a) - Q_{T-t+1}^{\pi^\star}(s, \pi^\star) \leq u$ for all actions $a$, time steps $t$ and states $s$ reachable by $\pi$. Then $\hat{\pi}$ satisfies $J_{\mathrm{C}}(\hat{\pi}) \leq J_{\mathrm{C}}(\pi^\star) + 3u\kappa\epsilon T$.*

Our bound is an improvement over that of behavior cloning if $\kappa$ is less than $\mathcal{O}(T)$. Note that DRIL does not require knowledge of $\kappa$. The quantity $\kappa$ is problem-dependent and depends on the

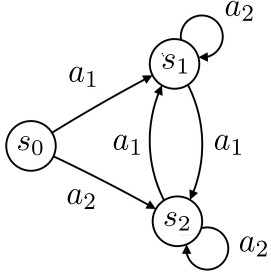

Figure 1: Example of a problem where behavioral cloning incurs quadratic regret.

environment dynamics, the expert policy and the policies in the learned ensemble. We next compute $\kappa$ exactly for a problem for which behavior cloning is known to perform poorly, and show that it is independent of $T$.

**Example 1.** *Consider the tabular MDP given in (Ross & Bagnell, 2010) as an example of a problem where behavioral cloning incurs quadratic regret, shown in Figure 1. There are 3 states $\mathcal{S} = (s_0, s_1, s_2)$ and two actions $(a_1, a_2)$. Each policy $\pi$ can be represented as a set of probabilities $\pi(a_1|s)$ for each state $s \in \mathcal{S}$ [2]. Assume the models in our ensemble are drawn from a posterior $p(\pi(a_1|s)|\mathcal{D})$ given by a Beta distribution with parameters $Beta(n_1 + 1, n_2 + 1)$ where $n_1, n_2$ are the number of times the pairs $(s, a_1)$ and $(s, a_2)$ occur, respectively, in the demonstration data $\mathcal{D}$. The agent always starts in $s_0$ and the expert's policy is given by $\pi^\star(a_1|s_0) = 1, \pi^\star(a_1|s_1) = 0, \pi^\star(a_1|s_2) = 1$. For any $(s, a)$ pair, the task cost is $C(s, a) = 0$ if $a = \pi^\star(s)$ and 1 otherwise. Here $d_\pi^\star = (\frac{1}{T}, \frac{T-1}{T}, 0)$. For any $\pi$, $d_\pi(s_0) = \frac{1}{T}$ and $d_\pi(s_1) \leq \frac{T-1}{T}$ due to the dynamics of the MDP, so $\frac{d_\pi(s)}{d_\pi^\star(s)} \leq 1$ for $s \in \{s_0, s_1\}$. Writing out $\alpha(\{s_0, s_1\})$, we get: $\alpha(\{s_0, s_1\}) = \max_{\pi \in \Pi} \sup_{s \in \{s_0, s_1\}} \frac{d_\pi(s)}{d_\pi^\star(s)} \leq 1$.*

*Furthermore, since $s_2$ is never visited in the demonstration data, for each policy $\pi_i$ in the ensemble we have $\pi_i(a_1|s_2), \pi_i(a_2|s_2) \sim Beta(1, 1) = Uniform(0, 1)$. It follows that $\mathrm{Var}_{\pi \sim \Pi_E}(\pi(a|s_2))$ is approximately equal [3] to the variance of a uniform distribution over $[0, 1]$, i.e. $\frac{1}{12}$. Therefore:*

$$\kappa = \min_{\mathcal{U} \subseteq \mathcal{S}} \frac{\alpha(\mathcal{U})}{\beta(\mathcal{U})} \leq \frac{\alpha(\{s_0, s_1\})}{\beta(\{s_0, s_1\})} \lesssim \frac{1}{\frac{1}{12}} = 12$$

*Applying our result from Theorem 3, we see that our algorithm obtains an $\mathcal{O}(\epsilon T)$ regret bound on this problem, in contrast to the $\mathcal{O}(\epsilon T^2)$ regret of behavioral cloning[4].*

## 5 RELATED WORK

The idea of learning through imitation dates back at least to the work of (Pomerleau, 1989), who trained a neural network to imitate the steering actions of a human driver using images as input. The problem of covariate shift was already observed, as the author notes: "the network must not solely be shown examples of accurate driving, but also how to recover once a mistake has been made".

This issue was formalized in the work of (Ross & Bagnell, 2010), who on one hand proved an $\mathcal{O}(\epsilon T^2)$ regret bound, and on the other hand provided an example showing this bound is tight. The subsequent work (Ross et al., 2011) proposed the DAGGER algorithm which obtains linear regret, provided the agent can both interact with the environment, and query the expert policy. Our approach also requires environment interaction, but importantly does not need to query the expert. Also of

---

[2]Note that $\pi(a_2|s) = 1 - \pi(a_1|s)$.

[3]Via Hoeffding's inequality, with probability $1 - \delta$ the two differ by at most $\mathcal{O}(\sqrt{\log(1/\delta)/|\Pi_E|})$.

[4]Observe that a policy with $\pi(a_1|s_0) = 1 - \epsilon T, \pi(a_2|s_1) = \epsilon T, \pi(a_2|s_2) = 1$ has a behavioral cloning cost of $\epsilon$ but a regret of $\epsilon T^2$.

note is the work of (Venkatraman et al., 2015), which extended DAGGER to time series prediction problems by using the true targets as expert corrections.

Imitation learning has been used within the context of modern RL to help improve sample efficiency (Chang et al., 2015; Ross & Bagnell, 2014; Sun et al., 2017; Hester et al., 2018; Le et al., 2018; Cheng & Boots, 2018) or overcome exploration (Nair et al., 2017). These settings assume the reward is known and that the policies can then be fine-tuned with reinforcement learning. In this case, covariate shift is less of an issue since it can be corrected using the reinforcement signal.

The work of (Luo et al., 2019) also proposed a method to address the covariate shift problem when learning from demonstrations when the reward is known, by conservatively extrapolating the value function outside the training distribution using negative sampling. This addresses a different setting from ours, and requires generating plausible states which are off the manifold of training data, which may be challenging when the states are high dimensional such as images. The work of (Reddy et al., 2019) proposed to treat imitation learning within the Q-learning framework, setting a positive reward for all transitions inside the demonstration data and zero reward for all other transitions in the replay buffer. This rewards the agent for repeating (or returning to) the expert's transitions. The work of (Sasaki et al., 2019) also incorporates a mechanism for reducing covariate shift by fitting a Q-function that classifies whether the demonstration states are reachable from the current state. Random Expert Distillation (Wang et al., 2019) uses Random Network Distillation (RND) (Burda et al., 2019) to estimate the support of the expert's distribution in state-action space, and minimizes an RL cost designed to guide the agent towards the expert's support. This is related to our method, but differs in that it minimizes the RND prediction error rather than the ensemble variance and does not include a behavior cloning cost. The behavior cloning cost is essential to our theoretical results and avoids certain failure modes, see Appendix B for more discussion.

Generative Adversarial Imitation Learning (GAIL) (Ho & Ermon, 2016) is a state-of-the-art algorithm which addresses the same setting as ours. It operates by training a discriminator network to distinguish expert states from states generated by the current policy, and the negative output of the discriminator is used as a reward signal to train the policy. The motivation is that states which are outside the training distribution will be assigned a low reward while states which are close to it will be assigned a high reward. This encourages the policy to return to the expert distribution if it strays away from it. However, the adversarial training procedure means that the reward function is changing over time, which can make the algorithm unstable or difficult to tune. In contrast, our approach uses a simple fixed reward function. We include comparisons to GAIL in our experiments.

Using disagreement between models in an ensemble to represent uncertainty has recently been explored in several contexts. The works of (Shyam et al., 2018; Pathak et al., 2019; Henaff, 2019) used disagreement between different dynamics models to drive exploration in the context of model-based RL. Conversely, (Henaff et al., 2019) used variance across different dropout masks to prevent policies from exploiting error in dynamics models. Ensembles have also been used to represent uncertainty over Q-values in model-free RL in order to encourage exploration (Osband et al., 2016). Within the context of imitation learning, the work of (Menda et al., 2018) used the variance of the ensemble together with the DAGGER algorithm to decide when to query the expert demonstrator to minimize unsafe situations. Here, we use disagreement between different policies trained on demonstration data to address covariate shift in the context of imitation learning.

## 6 EXPERIMENTS

### 6.1 TABULAR MDPS

As a first experiment, we applied DRIL to the tabular MDP of (Ross & Bagnell, 2010) shown in Figure 1. We computed the posterior over the policy parameters given the demonstration data using a separate Beta distribution for each state $s$ with parameters determined by the number of times each action was performed in $s$. For behavior cloning, we sampled a single policy from this posterior. For DRIL, we sampled an ensemble of 5 policies and used their negative variance to define an additional reward function. We combined this with a reward which was the probability density function of a given state-action pair under the posterior distribution, which corresponds to the supervised learning loss, and used tabular Q-learning to optimize the sum of these two reward functions. This experiment

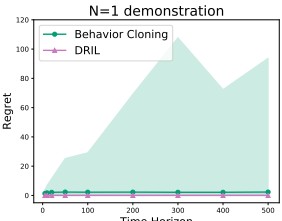 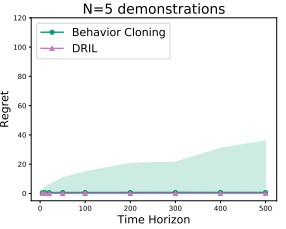 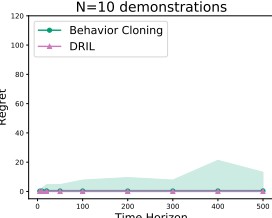

Figure 2: Results on tabular MDP from (Ross & Bagnell, 2010). Shaded region represents range between $5^{th}$ and $95^{th}$ quantiles, computed across 500 trials. Behavior cloning exhibits poor worst-case regret, whereas DRIL has low regret across all trials.

was repeated 500 times for time horizon lengths up to 500 and $N = 1, 5, 10$ expert demonstration trajectories.

Figure 2 shows plots of the regret over the 500 different trials across different time horizons. Although BC achieves good average performance, it exhibits poor worst-case performance with some trials incurring very high regret, especially when using fewer demonstrations. Our method has low regret across all trials, which stays close to constant independantly of the time horizon, even with a single demonstration. This performance is better than that suggested by our analysis, which showed a worst-case linear bound with respect to time horizon.

## 6.2 ATARI ENVIRONMENTS

We next evaluated our approach on six different Atari environments. We used pretrained PPO (Schulman et al., 2017) agents from the stable baselines repository (Hill et al., 2018) to generate $N = \{1, 3, 5, 10, 15, 20\}$ expert trajectories. We compared against two other methods: standard behavioral cloning (BC) and Generative Adversarial Imitation Learning (GAIL). Results are shown in Figure 3a. DRIL outperforms behavioral cloning across most environments and numbers of demonstrations, often by a substantial margin. In many cases, our method is able to match the expert's performance using a small number of trajectories. Figure 3b shows the evolution of the uncertainty cost and the policy reward throughout training. In all cases, the reward improves while the uncertainty cost decreases.

We were not able to obtain meaningful performance for GAIL on these domains, despite performing a hyperparameter search across learning rates for the policy and discriminator, and across different numbers of discriminator updates. We additionally experimented with clipping rewards in an effort to stabilize performance. These results are consistent with those of (Reddy et al., 2019), who also reported negative results when running GAIL on images. While improved performance might be possible with more sophisticated adversarial training techniques, we note that this contrasts with our method which uses a fixed reward function obtained through simple supervised learning.

In Appendix D we provide ablation experiments examining the effects of the cost function clipping and the role of the BC loss. We also compare the ensemble approach to a dropout-based approximation and show that DRIL works well in both cases.

## 6.3 CONTINUOUS CONTROL

We next report results of running our method on 6 different continuous control tasks from the PyBullet[5] and OpenAI Gym (Brockman et al., 2016) environments. We again used pretrained agents to generate expert demonstrations, and compared to Behavior Cloning and GAIL.

Results for all methods are shown in Figure 4. In these environments we found Behavior Cloning to be a much stronger baseline than for the Atari environments: in several tasks it was able to match expert performance using as little as 3 trajectories, suggesting that covariate shift may be less of an issue. Our method performs similarly to Behavior Cloning on most tasks, except on Walker2D, where it yields improved performance for $N = 1, 3, 5$ trajectories. GAIL performs

---

[5]https://github.com/bulletphysics/bullet3/tree/master/examples/pybullet/gym/pybullet_envs/examples

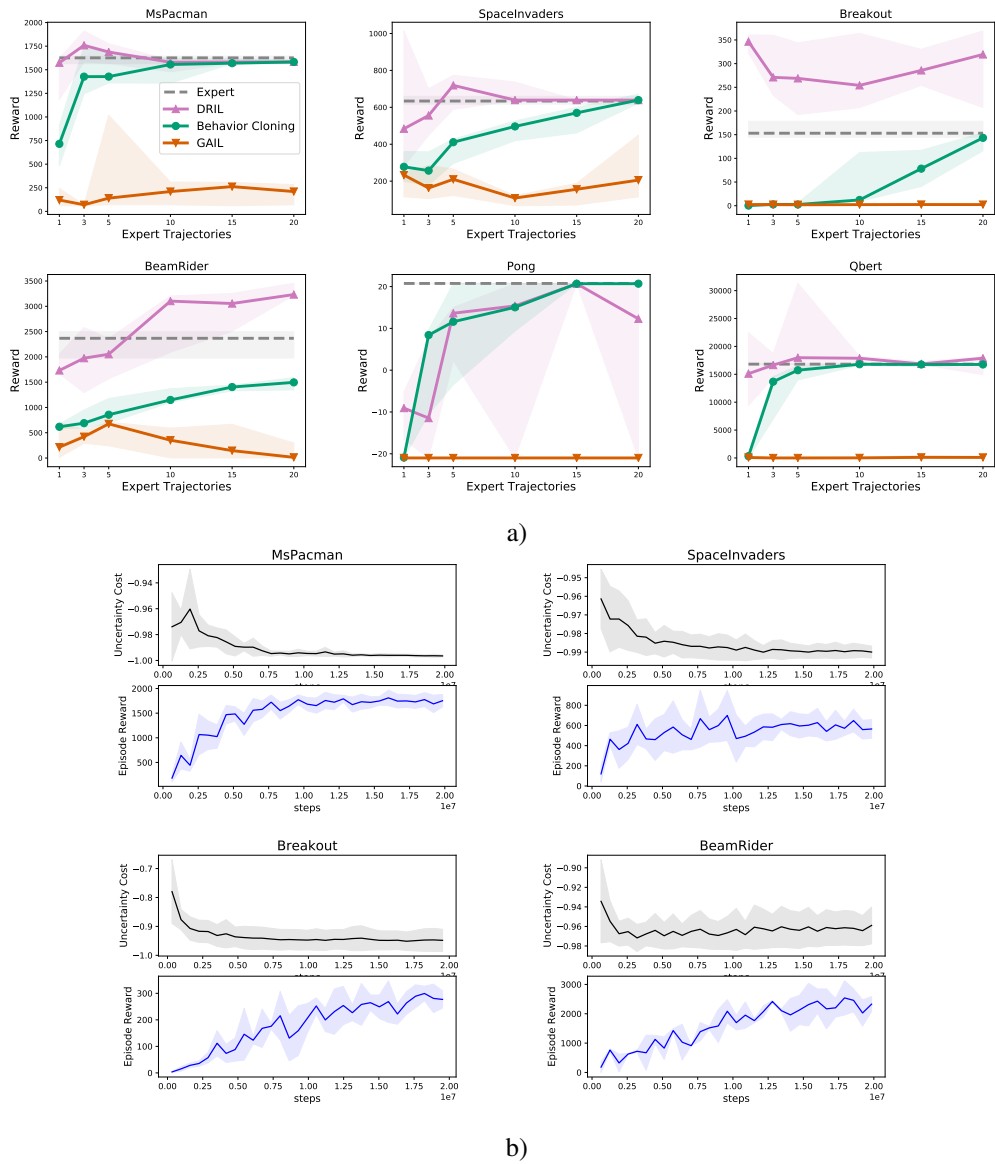

a)

b)

Figure 3: Results on Atari environments. a) Median final policy performance for different numbers of expert trajectories, taken over 4 seeds (shaded regions are min/max performance) b) Evolution of policy reward and uncertainty cost during training with $N = 3$ trajectories.

somewhat better than DRIL on HalfCheetah and Walker2D, but performs worse than both DRIL and BC on LunarLander and BipedalWalkerHardcore. The fact that DRIL is competitive across all tasks provides evidence of its robustness.

## 7 CONCLUSION

Addressing covariate shift has been a long-standing challenge in imitation learning. In this work, we have proposed a new method to address this problem by penalizing the disagreement between an ensemble of different policies trained on the demonstration data. Importantly, our method requires no additional labeling by an expert. Our experimental results demonstrate that DRIL can often match expert performance while using only a small number of trajectories across a wide array of tasks, ranging from tabular MDPs to pixel-based Atari games and continuous control tasks. On the theoretical side, we have shown that our algorithm can provably obtain a low regret bound for problems in which the $\kappa$ parameter is low.

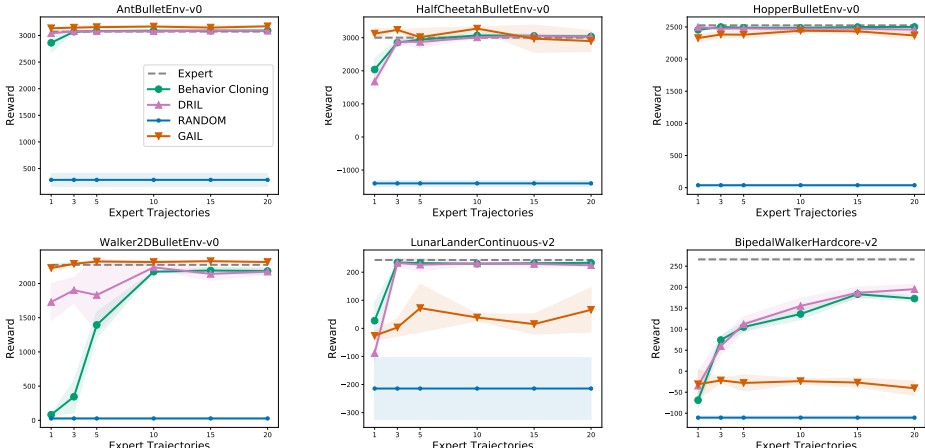

Figure 4: Results on continuous control tasks.

There are multiple directions for future work. On the theoretical side, characterizing the $\kappa$ parameter on a larger array of problems would help to better understand the settings where our method can expect to do well. Empirically, there are many other settings in structured prediction (Daumé et al., 2009) where covariate shift is an issue and where our method could be applied. For example, in dialogue and language modeling it is common for generated text to become progressively less coherent as errors push the model off the manifold it was trained on. Our method could potentially be used to fine-tune language or translation models (Cho et al., 2014; Welleck et al., 2019) after training by applying our uncertainty-based cost function to the generated text.

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

# A    PROOFS

**Lemma 1.** *For any $\pi \in \Pi$ we have $J_{\mathrm{exp}}(\pi) \leq \kappa J_{\mathrm{alg}}(\pi)$*

*Proof.* We will first show that for any $\pi \in \Pi$ and $\mathcal{U} \subseteq \mathcal{S}$, we have $J_{\mathrm{exp}}(\pi) \leq \frac{\alpha(\mathcal{U})}{\beta(\mathcal{U})} J_{\mathrm{alg}}(\pi)$. We can rewrite this as:

$$
\begin{aligned}
J_{\mathrm{exp}}(\pi) &= \mathbb{E}_{s \sim d_\pi} \Big[ \|\pi(\cdot|s) - \pi^\star(\cdot|s)\| \Big] \\
&= \mathbb{E}_{s \sim d_\pi} \Big[ \mathbb{I}(s \in \mathcal{U}) \|\pi(\cdot|s) - \pi^\star(\cdot|s)\| \Big] + \mathbb{E}_{s \sim d_\pi} \Big[ \mathbb{I}(s \notin \mathcal{U}) \|\pi(\cdot|s) - \pi^\star(\cdot|s)\| \Big]
\end{aligned}
$$

We begin by bounding the first term:

$$
\begin{aligned}
\mathbb{E}_{s \sim d_\pi} \Big[ \mathbb{I}(s \in \mathcal{U}) \|\pi(\cdot|s) - \pi^\star(\cdot|s)\| \Big] &= \sum_{s \in \mathcal{U}} d_\pi(s) \|\pi(\cdot|s) - \pi^\star(\cdot|s)\| \\
&= \sum_{s \in \mathcal{U}} \frac{d_\pi(s)}{d_{\pi^\star}(s)} d_{\pi^\star}(s) \|\pi(\cdot|s) - \pi^\star(\cdot|s)\| \\
&\leq \sum_{s \in \mathcal{U}} \underbrace{\left( \max_{\pi' \in \Pi} \sup_{s \in \mathcal{U}} \frac{d_{\pi'}(s)}{d_{\pi^\star}(s)} \right)}_{\alpha(\mathcal{U})} d_{\pi^\star}(s) \|\pi(\cdot|s) - \pi^\star(\cdot|s)\| \\
&= \alpha(\mathcal{U}) \sum_{s \in \mathcal{U}} d_{\pi^\star}(s) \|\pi(\cdot|s) - \pi^\star(\cdot|s)\| \\
&\leq \alpha(\mathcal{U}) \sum_{s \in \mathcal{S}} d_{\pi^\star}(s) \|\pi(\cdot|s) - \pi^\star(\cdot|s)\| \\
&= \alpha(\mathcal{U}) \mathbb{E}_{s \sim d_{\pi^\star}} \Big[ \|\pi(\cdot|s) - \pi^\star(\cdot|s)\| \Big] \\
&= \alpha(\mathcal{U}) J_{\mathrm{BC}}(\pi)
\end{aligned}
$$

We next bound the second term:

$$
\begin{aligned}
\mathbb{E}_{s \sim d_\pi} \Big[ \mathbb{I}(s \notin \mathcal{U}) \|\pi(\cdot|s) - \pi^\star(\cdot|s)\| \Big] &\leq \mathbb{E}_{s \sim d_\pi} \Big[ \mathbb{I}(s \notin \mathcal{U}) \Big] \\
&\leq \mathbb{E}_{s \sim d_\pi} \Big[ \mathbb{I}(s \notin \mathcal{U}) \frac{\min_{a \in \mathcal{A}} \mathrm{Var}_{\pi_i \sim \Pi_{\mathrm{E}}} [\pi_i(a|s)]}{\beta(\mathcal{U})} \Big] \\
&= \frac{1}{\beta(\mathcal{U})} \mathbb{E}_{s \sim d_\pi} \Big[ \mathbb{I}(s \notin \mathcal{U}) \sum_{a \in \mathcal{A}} \pi(a|s) \mathrm{Var}_{\pi_i \sim \Pi_{\mathrm{E}}} [\pi_i(a|s)] \Big] \\
&= \frac{1}{\beta(\mathcal{U})} \underbrace{\sum_{s \notin \mathcal{U}} d_\pi(s) \sum_{a \in \mathcal{A}} \pi(a|s) \mathrm{Var}_{\pi_i \sim \Pi_{\mathrm{E}}} [\pi_i(a|s)]}_{A(\pi)}
\end{aligned}
$$

Now observe we can decompose the RL cost as follows:

$$J_{\mathrm{U}}(\pi) = \mathbb{E}_{s \sim d_\pi, a \sim \pi(\cdot|s)}\left[\mathrm{Var}_{\pi_i \sim \Pi_{\mathrm{E}}} \pi_i(a|s)\right]$$

$$= \sum_s d_\pi(s) \sum_a \pi(a|s)\left[\mathrm{Var}_{\pi_i \sim \Pi_{\mathrm{E}}} \pi_i(a|s)\right]$$

$$= \underbrace{\sum_{s \in U} d_\pi(s) \sum_a \pi(a|s)\left[\mathrm{Var}_{\pi_i \sim \Pi_{\mathrm{E}}} \pi_i(a|s)\right]}_{B(\pi)} + \underbrace{\sum_{s \notin \mathcal{U}} d_\pi(s) \sum_a \pi(a|s)\left[\mathrm{Var}_{\pi_i \sim \Pi_{\mathrm{E}}} \pi_i(a|s)\right]}_{A(\pi)}$$

Putting these together, we get the following:

$$J_{\exp}(\pi) \le \alpha(\mathcal{U}) J_{\mathrm{BC}}(\pi) + \frac{1}{\beta(\mathcal{U})} A(\pi)$$

$$= \frac{\alpha(\mathcal{U})\beta(\mathcal{U})}{\beta(\mathcal{U})} J_{\mathrm{BC}}(\pi) + \frac{\alpha(\mathcal{U})}{\alpha(\mathcal{U})\beta(\mathcal{U})} A(\pi)$$

$$\le \frac{\alpha(\mathcal{U})}{\beta(\mathcal{U})} J_{\mathrm{BC}}(\pi) + \frac{\alpha(\mathcal{U})}{\beta(\mathcal{U})} A(\pi)$$

$$\le \frac{\alpha(\mathcal{U})}{\beta(\mathcal{U})}\left(J_{\mathrm{BC}}(\pi) + A(\pi)\right)$$

$$\le \frac{\alpha(\mathcal{U})}{\beta(\mathcal{U})}\left(J_{\mathrm{BC}}(\pi) + J_{\mathrm{U}}(\pi)\right)$$

$$= \frac{\alpha(\mathcal{U})}{\beta(\mathcal{U})} J_{\mathrm{alg}}(\pi)$$

Here we have used the fact that $\beta(\mathcal{U}) \le 1$ since $0 \le \pi(a|s) \le 1$ and $\alpha(\mathcal{U}) \ge \sup_{s \in \mathcal{U}} \frac{d_\pi^\star(s)}{d_\pi^\star(s)} = 1$ hence $\frac{1}{\alpha(\mathcal{U})} \le 1$. Taking the minimum over subsets $\mathcal{U} \subseteq \mathcal{S}$, we get $J_{\exp}(\pi) \le \kappa J_{\mathrm{alg}}(\pi)$.

$\square$

**Lemma 2.** $\min_{\pi \in \Pi} J_{\mathrm{alg}}(\pi) \le 2\epsilon$

*Proof.* Plugging the optimal policy into $J_{\mathrm{alg}}$, we get:

$$J_{\mathrm{alg}}(\pi^\star) = J_{\mathrm{BC}}(\pi^\star) + J_{\mathrm{U}}(\pi^\star)$$

$$= 0 + \mathbb{E}_{s \sim d_{\pi^\star}, a \sim \pi^\star(\cdot|s)}\left[\mathrm{Var}_{\pi_i \sim \Pi_{\mathrm{E}}}[\pi_i(a|s)]\right]$$

$$= \mathbb{E}_{s \sim d_{\pi^\star}, a \sim \pi^\star(\cdot|s)}\left[\frac{1}{E}\sum_{i=1}^{E}\left(\pi_i(a|s) - \bar{\pi}(a|s)\right)^2\right]$$

$$\le \mathbb{E}_{s \sim d_{\pi^\star}, a \sim \pi^\star(\cdot|s)}\left[\frac{1}{E}\sum_{i=1}^{E}\left(\pi_i(a|s) - \pi^\star(a|s)\right)^2 + \left(\bar{\pi}(a|s) - \pi^\star(a|s)\right)^2\right]$$

$$= \underbrace{\mathbb{E}_{s \sim d_{\pi^\star}, a \sim \pi^\star(\cdot|s)}\left[\frac{1}{E}\sum_{i=1}^{E}\left(\pi_i(a|s) - \pi^\star(a|s)\right)^2\right]}_{\mathrm{Term1}} + \underbrace{\mathbb{E}_{s \sim d_{\pi^\star}, a \sim \pi^\star(\cdot|s)}\left[\left(\bar{\pi}(a|s) - \pi^\star(a|s)\right)^2\right]}_{\mathrm{Term2}}$$

We will first bound Term 1:

$$\mathbb{E}_{s\sim d_{\pi^\star},a\sim\pi^\star(\cdot|s)}\Big[\frac{1}{E}\sum_{i=1}^{E}\Big(\pi_i(a|s)-\pi^\star(a|s)\Big)^2\Big] = \frac{1}{E}\mathbb{E}_{s\sim d_{\pi^\star}}\Big[\sum_{a\in\mathcal{A}}\pi^\star(a|s)\sum_{i=1}^{E}\Big(\pi_i(a|s)-\pi^\star(a|s)\Big)^2\Big]$$

$$\leq \frac{1}{E}\mathbb{E}_{s\sim d_{\pi^\star}}\Big[\sum_{a\in\mathcal{A}}\pi^\star(a|s)\sum_{i=1}^{E}\Big|\pi_i(a|s)-\pi^\star(a|s)\Big|\Big]$$

$$\leq \frac{1}{E}\mathbb{E}_{s\sim d_{\pi^\star}}\Big[\sum_{i=1}^{E}\sum_{a\in\mathcal{A}}\Big|\pi_i(a|s)-\pi^\star(a|s)\Big|\Big]$$

$$\leq \frac{1}{E}\sum_{i=1}^{E}\mathbb{E}_{s\sim d_{\pi^\star}}\Big[\|\pi_i(\cdot|s)-\pi^\star(\cdot|s)\|\Big]$$

$$\leq \frac{1}{E}\sum_{i=1}^{E}\epsilon$$

$$= \epsilon$$

We will next bound Term 2:

$$\mathbb{E}_{s\sim d_{\pi^\star},a\sim\pi^\star(\cdot|s)}\Big[\Big(\bar{\pi}(a|s)-\pi^\star(a|s)\Big)^2\Big] = \mathbb{E}_{s\sim d_{\pi^\star},a\sim\pi^\star(\cdot|s)}\Big[\Big(\pi^\star(a|s)-\frac{1}{E}\sum_{i=1}^{E}\pi_i(a|s)\Big)^2\Big]$$

$$= \mathbb{E}_{s\sim d_{\pi^\star},a\sim\pi^\star(\cdot|s)}\Big[\Big(\frac{1}{E}\sum_{i=1}^{E}\pi^\star(a|s)-\frac{1}{E}\sum_{i=1}^{E}\pi_i(a|s)\Big)^2\Big]$$

$$= \mathbb{E}_{s\sim d_{\pi^\star},a\sim\pi^\star(\cdot|s)}\Big[\Big(\frac{1}{E}\sum_{i=1}^{E}(\pi^\star(a|s)-\pi_i(a|s))\Big)^2\Big]$$

$$\leq \mathbb{E}_{s\sim d_{\pi^\star},a\sim\pi^\star(\cdot|s)}\Big[\frac{1}{E^2}E\sum_{i=1}^{E}\Big(\pi^\star(a|s)-\pi_i(a|s)\Big)^2\Big](\mathrm{Cauchy-Schwarz})$$

$$= \frac{1}{E}\sum_{i=1}^{E}\mathbb{E}_{s\sim d_{\pi^\star},a\sim\pi^\star(\cdot|s)}\Big[\Big(\pi^\star(a|s)-\pi_i(a|s)\Big)^2\Big]$$

$$\leq \frac{1}{E}\sum_{i=1}^{E}\mathbb{E}_{s\sim d_{\pi^\star},a\sim\pi^\star(\cdot|s)}\Big[\Big|\pi^\star(a|s)-\pi_i(a|s)\Big|\Big]$$

$$\leq \frac{1}{E}\sum_{i=1}^{E}\mathbb{E}_{s\sim d_{\pi^\star}}\Big[\|\pi^\star(\cdot|s)-\pi_i(\cdot|s)\|\Big]$$

$$= \frac{1}{E}\sum_{i=1}^{E}J_{\mathrm{BC}}(\pi_i)$$

$$\leq \epsilon$$

The last step follows from our optimization oracle assumption: $0 \leq \min_{\pi\in\Pi} J_{\mathrm{BC}}(\pi) \leq J_{\mathrm{BC}}(\pi^\star) = 0$, hence $J_{\mathrm{BC}}(\pi_i) \leq 0 + \epsilon = \epsilon$. Combining the bounds on the two terms, we get $J_{\mathrm{alg}}(\pi^\star) \leq 2\epsilon$. Since $\pi^\star \in \Pi$, the result follows.

$$\square$$

**Theorem 1.** *Let $\hat{\pi}$ be the result of minimizing $J_{\mathrm{alg}}$ using our optimization oracle, and assume that $Q^{\pi^\star}_{T-t+1}(s,a) - Q^{\pi^\star}_{T-t+1}(s,\pi^\star) \leq u$ for all $a \in \mathcal{A}, t \in \{1,2,...,T\}, d^t_\pi(s) > 0$. Then $\hat{\pi}$ satisfies $J(\hat{\pi}) \leq J(\pi^\star) + 3u\kappa\epsilon T$.*

*Proof.* By our optimization oracle and Lemma 2, we have

$$
\begin{aligned}
J_{\mathrm{alg}}(\hat{\pi}) &\leq \min_{\pi \in \Pi} J_{\mathrm{alg}}(\pi) + \epsilon \\
&\leq 2\epsilon + \epsilon \\
&= 3\epsilon
\end{aligned}
$$

Combining with Lemma 1, we get:

$$
\begin{aligned}
J_{\mathrm{exp}}(\hat{\pi}) &\leq \kappa J_{\mathrm{alg}}(\hat{\pi}) \\
&\leq 3\kappa\epsilon
\end{aligned}
$$

Applying Theorem 1 from (Ross et al., 2011), we get $J(\hat{\pi}) \leq J(\pi^\star) + 3u\kappa\epsilon T$.

$\square$

## B  IMPORTANCE OF BEHAVIOR CLONING COST

The following example shows how minimizing the uncertainty cost alone without the BC cost can lead to highly sub-optimal policies if the demonstration data is generated by a stochastic policy which is only slightly suboptimal. Consider the following deterministic chain MDP:

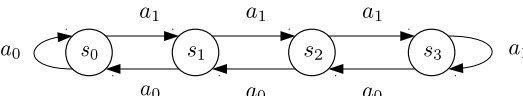

The agent always starts in $s_1$, and gets a reward of 1 in $s_3$ and 0 elsewhere. The optimal policy is given by:

$$
\begin{aligned}
\pi^\star(\cdot|s_0) &= (0,1) \\
\pi^\star(\cdot|s_1) &= (0,1) \\
\pi^\star(\cdot|s_2) &= (0,1) \\
\pi^\star(\cdot|s_3) &= (0,1)
\end{aligned}
$$

Assume the demonstration data is generated by the following policy, which is only slightly suboptimal:

$$
\begin{aligned}
\pi_{\mathrm{demo}}(\cdot|s_0) &= (0,1) \\
\pi_{\mathrm{demo}}(\cdot|s_1) &= (0,1) \\
\pi_{\mathrm{demo}}(\cdot|s_2) &= (0.1,0.9) \\
\pi_{\mathrm{demo}}(\cdot|s_3) &= (0,1)
\end{aligned}
$$

Let us assume realizability and perfect optimization for simplicity. If both transitions $(s_2,a_0)$ and $(s_2,a_1)$ appear in the demonstration data, then Random Expert Distillation (RED) will assign zero

cost to both transitions. If we do not use bootstrapped samples to train the ensemble, then DRIL without the BC cost (we will call this UO-DRIL for Uncertainty-Only DRIL) will also assign zero cost to both transitions since all models in the ensemble would recover the Bayes optimal solution given the demonstration data. If we are using bootstrapped samples, then the Bayes optimal solution for each bootstrapped sample may differ and thus the different policies in the ensemble might disagree in their predictions, although given enough demonstration data we would expect these differences (and thus the uncertainty cost) to be small.

Note also that since no samples at the state $s_0$ occur in the demonstration data, both RED and UO-DRIL will likely assign high uncertainty costs to state-action pairs at $(s_0, a_0)$, $(s_0, a_1)$ and thus avoid highly suboptimal policies which get stuck at $s_0$.

Now consider policies $\hat{\pi}_1, \hat{\pi}_2$ given by:

$$
\begin{aligned}
\hat{\pi}_1(\cdot|s_0) &= (0, 1) \\
\hat{\pi}_1(\cdot|s_1) &= (0, 1) \\
\hat{\pi}_1(\cdot|s_2) &= (1, 0) \\
\hat{\pi}_1(\cdot|s_3) &= (0, 1)
\end{aligned}
$$

and

$$
\begin{aligned}
\hat{\pi}_2(\cdot|s_0) &= (0, 1) \\
\hat{\pi}_2(\cdot|s_1) &= (0, 1) \\
\hat{\pi}_2(\cdot|s_2) &= (0.2, 0.8) \\
\hat{\pi}_2(\cdot|s_3) &= (0, 1)
\end{aligned}
$$

Both of these policies only visit state-action pairs which are visited by the demonstration policy. In the case described above, both RED and UO-DRIL will assign $\hat{\pi}_1$ and $\hat{\pi}_2$ similarly low costs. However, $\hat{\pi}_1$ will cycle forever between $s_1$ and $s_2$, never collecting reward, while $\hat{\pi}_2$ will with high probability reach $s_3$ and stay there, thus achieving high reward. This shows that minimizing the uncertainty cost alone does not necessarily distinguish between good and bad policies. However, $\hat{\pi}_1$ will incur a higher BC cost than $\hat{\pi}_2$, since $\hat{\pi}_2$ more closely matches the demonstration data at $s_2$. This shows that including the BC cost can be important for further disambiguating between policies which all stay within the distribution of the demonstration data, but have different behavior within that distribution.

## C    EXPERIMENTAL DETAILS

### C.1    ATARI ENVIRONMENTS

All behavior cloning models were trained to minimize the negative log-likelihood classification loss on the demonstration data for 500 epochs using Adam (Kingma & Ba, 2014) and a learning rate of $2.5 \cdot 10^{-4}$. We stopped training once the validation error did not improve for 20 epochs. For our method, we initially performed a hyperparameter search on Space Invaders over the values shown in Table 1

Table 1: Hyperparameters for DRIL

| Hyperparameter | Values Considered | Final Value |
|---|---|---|
| Policy Learning rate | $2.5 \cdot 10^{-2}, 2.5 \cdot 10^{-3}, 2.5 \cdot 10^{-4}$ | $2.5 \cdot 10^{-3}$ |
| Quantile cutoff | $0.8, 0.9, 0.95, 0.98$ | 0.98 |
| Number of supervised updates | $1, 5$ | 1 |
| Number of policies in ensemble | 5 | 5 |
| Gradient clipping | 0.1 | 0.1 |
| Entropy coefficient | 0.01 | 0.01 |
| Value loss coefficient | 0.5 | 0.5 |
| Number of steps | 128 | 128 |
| Parallel Environments | 16 | 16 |

We then chose the best values and kept those hyperparameters fixed for all other environments. All other A2C hyperparameters follow the default values in the repo (Kostrikov, 2018): policy networks consisted of 3-layer convolutional networks with $8 - 32 - 64$ feature maps followed by a single-layer MLP with 512 hidden units.

For GAIL, we used the implementation in (Kostrikov, 2018) and replaced the MLP discriminator by a CNN discriminator with the same architecture as the policy network. We initially performed a hyperparameter search on Breakout with 10 demonstrations over the values shown in Table 2. However, we did not find any hyperparameter configuration which performed better than behavioral cloning.

Table 2: Hyperparameters for GAIL

| Hyperparameter | Values Considered | Final Value |
|---|---|---|
| Policy Learning rate | $2.5 \cdot 10^{-2}, 2.5 \cdot 10^{-3}, 2.5 \cdot 10^{-4}$ | $2.5 \cdot 10^{-3}$ |
| Discriminator Learning rate | $2.5 \cdot 10^{-2}, 2.5 \cdot 10^{-3}, 2.5 \cdot 10^{-4}$ | $2.5 \cdot 10^{-3}$ |
| Number of discriminator updates | $1, 5, 10$ | 5 |
| Gradient clipping | 0.1 | 0.1 |
| Entropy coefficient | 0.01 | 0.01 |
| Value loss coefficient | 0.5 | 0.5 |
| Number of steps | 128 | 128 |
| Parallel Environments | 16 | 16 |

## C.2 CONTINUOUS CONTROL

All behavior cloning and ensemble models were trained to minimize the mean-squared error regression loss on the demonstration data for 500 epochs using Adam (Kingma & Ba, 2014) and a learning rate of $2.5 \cdot 10^{-4}$. Policy networks were 2-layer fully-connected MLPs with tanh activations and 64 hidden units.

Table 3: Hyperparameters (our method)

| Hyperparameter | Values Considered | Final Value |
|---|---|---|
| Policy Learning rate | $2.5 \cdot 10^{-3}, 2.5 \cdot^{1} 0 - 4, 1 \cdot 10^{-4}, 5 \cdot 10^{-5}$ | $2.5 \cdot 10^{-5}$ |
| Quantile cutoff | 0.98 | 0.98 |
| Number of supervised updates | 1 | 1 |
| Number of policies in ensemble | 5 | 5 |
| Gradient clipping | 0.1 | 0.1 |
| Entropy coefficient | 0.01 | 0.01 |
| Value loss coefficient | 0.5 | 0.5 |
| Number of steps | 128 | 128 |
| Parallel Environments | 16 | 16 |

# D   ABLATION EXPERIMENTS

In this section we provide ablation experiments examining the effects of the cost function clipping and the role of the BC loss. We also compare the ensemble approach to a dropout-based approximation and show that DRIL works well in both cases.

Table 4: Ablation Experiments with 3 expert trajectories

| Environment | SpaceInvaders | Breakout | BeamRider |
|---|---|---|---|
| DRIL (ensemble) | 555.7 | 286.7 | 2033.4 |
| DRIL (dropout) | 581.4 | 205.4 | 2124.5 |
| DRIL (raw cost) | 421.8 | 70.9 | 1265.5 |
| DRIL (no BC cost) | 102.1 | 78.3 | 538.4 |
| BC | 257.0 | 2.7 | 689.7 |

Results are shown in Figure 4. First, switching from the clipped cost in $\{-1, +1\}$ to the the the raw cost causes a drop in performance. One explanation may be that since the raw costs are always positive (which corresponds to a reward which is always negative), the agent may learn to terminate the episode early in order to minimize the total cost incurred. Using a cost/reward which has both positive and negative values avoids this behavior.

Second, optimizing the pure BC cost performs better than the pure uncertainty cost for some environments (SpaceInvaders, BeamRider) while optimizing the pure uncertainty cost performs better than BC in Breakout. DRIL, which optimizes both, has robust performance and performs the best over all environments.

For the dropout approximation we trained a single policy network with a dropout rate of $0.1$ applied to all layers except the last, and estimated the variance for each state-action pair using $5$ different dropout masks. Similarly to the ensemble approach, we computed the $98^{\text{th}}$ quantile of the variance on the demonstration data and used this value in our clipped cost. MC-dropout performs similarly to the ensembling approach, which shows that our method can be paired with different approaches to posterior estimation.

