# OpenReview forum: "Disagreement-Regularized Imitation Learning"
_ICLR.cc/2020/Conference — Accept (Spotlight)_

### Official Review · AnonReviewer3 · 2019-10-18
**Official Blind Review #3**

**Rating:** 8

**Review:**

* Summary:
The paper aims to address the covariate shift issue of behavior cloning (BC). The main idea of the paper is to learn a policy by minimizing a BC loss and an uncertainty loss. This uncertainty loss is defined as a variance of a policy posterior given by demonstration. To approximate this posterior, the paper uses an ensemble approach, where an ensemble of policies is learned from demonstrations. This approach leads to a method called disagreement-regularized imitation learning (DRIL). The paper proofs for a tabular setting that DRIL has a linear regret bound in terms of the horizon, which is better than that of BC which has a quadratic regret bound. Empirical evaluation shows that DRIL outperforms BC in both discrete and continuous control tasks, and it outperforms GAIL in discrete control tasks.

* General comments:
The paper proposes a simple but effective method to address the important issue of covariate shift. The method performs well empirically and has a theoretical support (although only for a tabular setting). While there are some issues (see below), this is a good paper. I vote for acceptance.

* Major comments and questions:
- Accuracy of posterior approximation via ensemble.
It is unclear whether the posterior approximated from ensemble is accurate. More specifically, these ensemble policies are trained using BC loss. Under a limited amount of data (where BC fails), these policies would also fail and are inaccurate. Therefore, it should not be expected that a posterior from these inaccurate policies is accurate. Have the authors measure or analyze accuracy of these policies or that of the posterior? This important point is not mentioned or analyzed in the paper.

- Alternative approaches to posterior approximation and uncertainty computation.
There are other approaches to obtain a posterior besides the ensemble approach, e.g., Bayesian neural networks. Such alternatives were not mentioned in the paper. Also, there are other quantities for measuring uncertainty besides the variance such as the entropy. These approaches and quantities have different pros and cons and they should be discussed in the paper.

- Sample complexity in terms of environment interactions.
The sample complexity in terms of environment interactions is an important criterion for IL. I suggest the authors to include this criterion in the experiments.

* Minor questions:
- Why does the minibatch size is only 4 in the experiments for all methods. This is clearly too small for a reasonable training of deep networks. Is this a typo?

- It is strange to not evaluate GAIL in the continuous control experiments, since GAIL was originally evaluated in these domains. I strongly suggest the authors to evaluate GAIL (and perhaps stronger methods such as VAIL (Peng et al., 2019)) in the continuous control experiments.

---After reading authors' response---
I have read the authors' response and other reviews. The authors addressed my comments in the response and the updated paper. I keep the same rating and recommend acceptance.


**Experience Assessment:**

I have published one or two papers in this area.

**Review Assessment: Checking Correctness Of Derivations And Theory:**

I assessed the sensibility of the derivations and theory.

**Review Assessment: Checking Correctness Of Experiments:**

I assessed the sensibility of the experiments.

**Review Assessment: Thoroughness In Paper Reading:**

I read the paper at least twice and used my best judgement in assessing the paper.

---

> ### Author Response · Authors · 2019-11-15
> **Response to Reviewer #3**
>
> Thank you for the review. To address the questions/comments:
>
> Major questions/comments:
>
> - It is true that we should not expect the policies in the ensemble to perform better than BC, since they are trained on the same limited data. However, the motivation is that even though they may make errors, the errors they make are likely to be different from each other. For example, if we look at several functions sampled from a Gaussian process posterior, these will tend to agree on the training data, but can look very different (both from the true function and each other) outside of the training data. Therefore we do not care so much about the quality of the ensemble policies (measured by how they would perform in the environment), but rather whether they exhibit low variance on the training data and higher variance off of it.
>
> - We have updated the text to mention that other methods for posterior approximation are also possible (Bayes by Backprop, MC-dropout), and added additional experiments comparing the ensemble approach to MC-dropout in Appendix D2. It turns out that MC-dropout also works well, similarly to the ensemble method. This shows that our approach is not specific to the ensemble method which we use in most of our experiments.
>
> - We have added the number of environment steps to the curves in Figure 2b, which shows the sample complexity. Note that since we use A2C as an RL optimizer in our experiments, we are not particularly sample efficient in terms of environment steps. Our general method is agnostic to the RL optimizer though, so more sample-efficient RL methods (such as model-based methods or others which reuse data more efficiently) could in principle be used as well.
>
>
> Minor questions:
>
> - Minibatch 4 was a typo, thanks for catching that. We use 16 parallel environments for A2C and have added this to the experiment details.
>
> - We initially did not include GAIL in the continuous control experiments because there was not much headroom for improvement over BC. We will add these experiments for the next update.

---

### Official Review · AnonReviewer1 · 2019-10-23
**Official Blind Review #1**

**Rating:** 8

**Review:**

The paper proposes an imitation learning algorithm that combines behavioral cloning with a regularizer that encourages the agent to visit states similar to the demonstrated states. The key idea is to use ensemble disagreement to approximate uncertainty, and use RL to train the imitation agent to visit states in which an ensemble of cloned imitation policies is least uncertain about which action the expert would take. Experiments on image-based Atari games show that the proposed method significantly outperforms BC and GAIL baselines in three games, and performs comparably or slightly better than the baselines in the remaining three games.

Overall, I enjoyed reading this paper. It proposes a relatively simple imitation method with compelling empirical results.

One minor comment: on page 15, the sentence "We initially performed a hyperparameter search on Breakout with 10 demonstrations over the following values: " ends in a blank space, without actually providing any hyperparameter values. It would be nice if you could actually include those values, or at least how many different values were searched.

Thank you for addressing the comments about related work in an earlier thread (https://openreview.net/forum?id=rkgbYyHtwB&noteId=S1lv4r5qvS). Two follow-ups:
 - The chain MDP example clearly illustrates why including the BC cost is important, and how DRIL differs from support estimation methods like RED. Thank you for the clarification.
 - The focus of Sasaki et al. is on reducing the number of environment interactions, but their proposed method also addresses covariate shift: it fits a Q function that classifies whether the demonstration states are reachable from the current state, and thus encourages the agent to return to demonstrated states.

**Experience Assessment:**

I have read many papers in this area.

**Review Assessment: Checking Correctness Of Derivations And Theory:**

I assessed the sensibility of the derivations and theory.

**Review Assessment: Checking Correctness Of Experiments:**

I assessed the sensibility of the experiments.

**Review Assessment: Thoroughness In Paper Reading:**

I read the paper at least twice and used my best judgement in assessing the paper.

---

> ### Author Response · Authors · 2019-11-15
> **Response to Reviewer #1**
>
> Thank you for the encouraging comments and we are glad you enjoyed reading the paper. Regarding the GAIL hyperparameters: this was a formatting issue and we have changed the text to refer to Table 2 where the GAIL hyperparameters are listed. We have also added the chain MDP example to the appendix and added references discussed in the previous comment thread.

---

### Official Review · AnonReviewer2 · 2019-10-24
**Official Blind Review #2**

**Rating:** 6

**Review:**

Summary of what the paper claims and contributes
---
This paper proposes a new interactive imitation learning algorithm to address the covariate shift problem in imitation learning. It explicitly seeks to avoid settings interactive expert feedback (e.g. DAgger). The method is straightforward: 1. First, learn an ensemble of policies via KL-based Behavior Cloning 2. Then, learn a new policy via a new objective that combines the original Behavior Cloning objective with a "disagreement" loss, formed by computing the expected variance of the ensemble evaluated on state-action trajectories under the new policy. The intuition for the method is that by learning an ensemble, it will have low variance on in-distribution demonstration data, and high variance on out-of-distribution other data; by encouraging the policy to seek regions of low variance, it should result in a policy that more closely matches the demonstrator's state visitation distribution than Behavior-Cloning alone. Analysis in the discrete finite case shows that the algorithm achieves regret linear in \kappa*T, where \kappa is an environment- and expert-dependent constant. The analysis is instantiated for a simple MDP, and experiments comparing their algorithm on this restricted environment provide some evidence that the bound is achievable in practice.

Further experiments on a variety of Atari environments and continuous-control tasks from OpenAI Gym also 1) demonstrates that their algorithm outperforms Behavior Cloning in these settings 2) usually approaches expert performance with a small number of demonstrations, and 3) also shows that the uncertainty cost improves over time, indicating the final policy learns to visit states where the ensemble agrees, and that while doing so, improves performance on the underlying task.

Evaluation
---
>Originality:
Are the tasks or methods new?
The method is new.

Is the work a novel combination of well-known techniques?
Yes.

Is it clear how this work differs from previous contributions?
Yes.

Is related work adequately cited?
There is some missing discussion of related works:
1. EnsembleDAgger (Menda 2018) also uses the variance of ensembles in Imitation Learning, but instead of using it to regularize on-policy learning, it uses it as an improved decision criterion by which to query an expert demonstrator.
2. Data as Demonstrator (Venkatraman 2015) uses on-policy learning to create "corrections" of time-series models (See their Fig 1), which is similar to this paper's intuition of seeking to push the learner back to places that are in-distribution of the expert demonstrations. That paper also achieves a linear regret bound under some assumptions.

>Quality:
Is the submission technically sound?
Mostly, although there are some issues:
1. Step 9 of the algorithm is ambiguous. What is the distribution of on-policy data that is fed into the cost? E.g. how many rollouts from the policy are collected?
2. Why is the clipped cost negative, as opposed to 0?
3. Why was a clipped cost used at all? This cost is different from that used in the theoretical analysis. Some justification and discussion is needed for why the new cost was used, and whether the analysis still applies when it's used.
4. Throughout most of the paper, p(\pi | \mathcal D) represents the model ensemble. However, no discussion was dedicated to what we should expect this distribution to look like in theory and in practice. It depends on how the ensemble is constructed / learned. A degenerate case would be if all models in the ensemble converged to the same local optima, in which case they would agree everywhere, nullifying the cost penalty. Discussion of what properties this distribution must satisfy is missing. It probably needs full support over the space of policies such that the optimal policy is nearly realizable (within \epsilon)?
5. \kappa is overloaded: A. it's used as a function B. it's used as the optimal value of that same function. Consider using different notation for one of the, e.g. \kappa^* for the optimum, or \gamma for the function. Furthermore, it might help to make \kappa's dependencies clearer, which would help illustrate its independence of T.
6. Example 1: the fact that the policy always starts at s_1 is missing from the description (at least, an equivalent assumption is made in Ross 2010)
7. Example 1: it's not clear that setting \mathcal U = \{s_1, s_2\} achieves the optimum of \kappa(\mathcal U). Discussion of this aspect is needed.
8. Example 1: The statement that the variance is equivalent to the variance of the uniform distribution seems to be a strong assumption about p(\pi | \mathcal D). This missing assumption is related to point 4. I mentioned above^
9. The paper is missing discussion for why the analysis would not immediately extend to continuous state and action spaces.

Are claims well supported by theoretical analysis or experimental results?
Yes, although the experimental results would be made stronger if related approaches were considered, e.g. Reddy 2019. Right now, there's just a single method of comparison -- BC.

Is this a complete piece of work or work in progress?
Seems complete.

Are the authors careful and honest about evaluating both the strengths and weaknesses of their work?
I believe so -- noting that BC ended up performing similar in environments where there is less drift was a good addition.

>Clarity:
Is the submission clearly written?
Yes.

Is it well organized?
Yes.

Does it adequately inform the reader?
Yes.

>Significance:
Are the results important?
Yes.

Are others (researchers or practitioners) likely to use the ideas or build on them?
Yes.

Does the submission address a difficult task in a better way than previous work?
Yes.

Does it advance the state of the art in a demonstrable way?
Yes.

Does it provide unique data, unique conclusions about existing data, or a unique theoretical or experimental approach?
Unique theoretical approach.

Additional feedback
---
Sec 3: "The threshold q defines a normal range of uncertainty based on the demonstration data, and values outside of this range incur a negative cost". The logic of this statement is confusing. 1. It's not clear what "outside" means from the sentence alone (i.e. it should be "above"). 2. A single value doesn't define a range (i.e. state the lower value is 0).

Sec 4.1: "high density" -> "high mass"

It would help to have a diagram of \mathcal U, \mathcal S - \mathcal U, \alpha, \beta, \kappa.

It would be clearer if set notation was used for the complement of \mathcal U, rather than \beta's definition of s\notin \mathcal U.

Example 1: citation should be Ross 2010, not Ross 2011.

Example 1 has different notation than in Ross 2010 (consider changing to match)

It's possible that copying a model from the ensemble and fine-tuning it with the loss would yield a faster Algorithm (1). Would this work? What do the training curves (i.e. like the plots in Fig 3b) look like in that case?

Why does the breakout DRIL agent outperform the expert?

Mention that Pinkser's inequality yields the KL bound on total variation.



**Experience Assessment:**

I have published one or two papers in this area.

**Review Assessment: Checking Correctness Of Derivations And Theory:**

I assessed the sensibility of the derivations and theory.

**Review Assessment: Checking Correctness Of Experiments:**

I assessed the sensibility of the experiments.

**Review Assessment: Thoroughness In Paper Reading:**

I read the paper at least twice and used my best judgement in assessing the paper.

---

> ### Author Response · Authors · 2019-11-15
> **Response to Reviewer #2**
>
> Thank you for the detailed review and suggestions for improving the paper. We have made the following changes in response:
>
> - We have added references to the two suggested related works (Menda 2018 and Venkatraman 2015).
>
> 1. We have clarified that Step 9 of the algorithm optimizes the expected clipped cost under the current policy. In our experiments we use A2C, which estimates the expected cost using rollouts from multiple parallel actors all sharing the same policy (16 in our case, we have added this to the experiment details in Appendix C).
>
> 2-3. We have added ablation experiments in Appendix D1 showing the effect of the different choices for the cost clipping (negative vs. 0, not clipping at all). Having the range of the cost (or reward) include negative and positive values has a large impact on performance. We believe the reason is that if the cost is always positive (or reward is always negative), then an easy way to minimize the cost (or maximize reward) is for the agent to terminate the episode early. Some environments such as Mountain Car are in fact designed this way: all rewards are negative, and the optimal policy is to reach the goal (and thus terminate the episode) as soon as possible. In other environments however, terminating early is highly suboptimal (i.e. the agent dies and cannot collect any more reward). Including both positive and negative costs helps to avoid these issues.
>
> 4. We train the different models in the ensemble starting from different initializations and using different bootstrap samples of the demonstration data (we have made this more clear in the text). While it is true that the degenerate case of all models converging to the same solution could potentially occur, our experiments and other works which successfully use ensembles for posterior approximation (mentioned in related work) suggest that this is rare in practice. We have also added experiments in Appendix D2 comparing ensembles to MC-dropout for posterior approximation, and found that dropout also works well - this shows that our approach is not specifically tied to the ensemble method.
>
> 5. We have changed notation to use \kappa^* for the optimum.
>
> 6. We have specified the agent's start state, and changed the notation to be consistent with the original work.
>
> 7. Our goal is to show that \kappa^* is upper bounded by a constant independent of T, which translates into a better regret bound than BC when T becomes large. Since \kappa^* is the minimum of \kappa(U) for all subsets U of S, showing that \kappa(U) is upper bounded by a constant for some U means that \kappa^* is also. We have clarified this in the example.
>
> 8. In Example 1, we have specified that we are using a Beta distribution to represent the posterior, whose parameters are determined by the state-action counts in the demonstration data (Beta/Dirichlets are standard choices for binomial/categorical distributions). For the state s_2 which is never visited, the Beta distribution becomes equivalent to a uniform distribution, which is where we get our value of the variance from.
>
> 9. Most of the derivations do carry over to the continuous setting, but there are two steps in the last part of the proof of Lemma 1 that use properties of discrete states/actions: that \alpha(U) >= 1, and that \pi(a | s) \leq 1 (note that for continuous actions, densities can become arbitrarily peaked so the last bound, which was used to bound \beta(U), does not hold). We are currently working on the continuous case but our current results are for the tabular case.
>
>
> Additional feedback:
>
> We have made a number of additional changes: fixing the citation, changing notation in the example, changing density to mass, added mention of Pinsker's inequality and changed the wording regarding the q threshold.

---

### Public Comment · ~Siddharth_Reddy1 · 2019-09-26
**Related work**

Thank you for the interesting paper! I have two comments about related work.

There are two prior methods – Random Expert Distillation (RED) by Wang et al. [1], and the implicit IRL method in Sasaki et al. [2] – that aren’t mentioned in the paper, but are similar to the proposed method. In particular, the proposed method seems to take a similar approach to RED, except that it uses ensemble disagreement instead of random network distillation for density estimation of the demonstrations. It would be nice to discuss how the proposed method relates to the prior work.

The discussion of one of the prior methods – SQIL by Reddy et al. – mischaracterizes how SQIL works. The first paragraph on page 6 claims that SQIL requires careful reward decay and does not use a fixed reward function. In fact, SQIL uses a fixed reward function (r=+1 for demonstrations, r=0 for everything else), and does not modify or decay the rewards over time. It would be nice to adjust how SQIL is positioned in the related work. In my opinion, the proposed method differs from SQIL in that it uses a fixed reward function that is potentially less sparse and potentially easier to train the imitation agent with via RL.

Again, thank you for the interesting work. I look forward to seeing how the paper evolves, and hope that others working on imitation learning give it a read.

[1] http://proceedings.mlr.press/v97/wang19d/wang19d.pdf
[2] https://openreview.net/forum?id=BkN5UoAqF7

---

> ### Author Response · Authors · 2019-10-06
> **Addressing Related work comments:**
>
> Thank you for the helpful pointers to the related work. The Random Expert Distillation (RED) method by Wang et al [1] is indeed relevant and we will include a discussion in the updated paper. Both RED and our method use an uncertainty measure derived from the demonstration data as a cost function which is minimized through RL, and which is designed to guide the policy back towards the demonstration data. There are two main differences between RED and our method: i) we use the variance of an ensemble as a measure of uncertainty, rather than random network distillation [3] and ii) we include a supervised behaviour cloning (BC) cost in addition to the uncertainty cost.
>
> Including the BC cost is actually quite important for our theoretical results. In Lemma 1, J_{exp}(\pi) is broken up into two terms, one of which is bounded by the BC cost (scaled by alpha(U)) and one of which is bounded by the uncertainty cost (scaled by 1/beta(U)). By minimizing both of these costs, we minimize J_{exp}(\pi) (scaled by kappa), which in turn translates into a regret bound.
>
> The following example shows that minimizing the uncertainty cost alone without the BC cost can lead to highly sub-optimal policies if the demonstration data is generated by a stochastic policy which is only slightly suboptimal. Consider the following deterministic chain MDP:
>
> s0 <---> s1 <---> s2 <---> s3
>
> Say the agent always starts in s1, and gets a reward of 1 in s3 and 0 elsewhere, and there are 2 actions: left and right (in s3, going right keeps the agent at s3, in s0 going left keeps the agent at s0).
>
> Assume the demonstration data is generated by a policy defined as follows:
> - in s0, go right with probability 1
> - in s1, go right with probability 1
> - in s2, go right with probability 0.9, left with probability 0.1
> - in s3, go right (i.e. stay at s3) with probability 1.
>
> If both transitions (s2, right) and (s2, left) appear in the demonstration data, then (assuming realizability) RED will assign the same cost to both transitions. This means that a policy which cycles forever between s1 and s2 (always going left at s2, and never collecting reward) will have the same cost as a policy which goes right at s2 and then stays at s3 (thus collecting lots of reward). If we include a BC cost however, the policy will learn to assign a higher probability to going right at s2 and end up collecting reward. For both RED and our method, if we are realizable and optimization can be performed exactly, then the uncertainty cost will be set to zero for all transitions appearing in the demonstration data, regardless of their relative frequency. However, this problem can be avoided by combining the BC cost with the uncertainty cost.
>
> The method of Sasaki et. al [2] is interesting and we will include a reference in related work. The focus of their work is somewhat different, i.e. reducing the number of environment interactions rather than addressing covariate shift.
>
> Thank you for the recommendations regarding the description of SQIL, we will include them when we update the paper.
>
> [1] http://proceedings.mlr.press/v97/wang19d/wang19d.pdf
> [2] https://openreview.net/forum?id=BkN5UoAqF7
> [3] https://arxiv.org/pdf/1810.12894.pdf

---

### Author Response · Authors · 2019-11-15
**Updates**

We have made a number of updates to the paper in response to the comments, please see our answers below. We have also changed the colors of the plots to be more color-blind friendly.

---

### Decision · Program_Chairs · 2019-12-19

**Decision:**

Accept (Spotlight)

**Comment:**

This paper presents an approach for interactive imitation learning while avoiding an adversarial optimization by using ensembles. The reviewers agreed that the contributions were significant and the results were compelling. Hence, the paper should be accepted.

---

> ### Public Comment · ~Akshay_Krishnamurthy1 · 2020-03-04
> **Minor comment**
>
> Hi -- I just want to point out that this paper is _not_ studying interactive imitation learning. It is considering the non-interactive setting, where we cannot query the expert, but we do see the expert's actions.

---

> > ### Comment · Area_Chair1 · 2020-04-28
> > **Response**
> >
> > To clarify, it is interactive imitation learning in the sense that the algorithm can collect additional data in the environment. This is in contrast to supervised behavior cloning algorithms that only use demonstrations and no additional environment roll-outs.